

# Whole-genome single-nucleotide polymorphism (SNP) marker discovery and association analysis with the eicosapentaenoic acid (EPA) and docosahexaenoic acid (DHA) content in *Larimichthys crocea*

Shijun Xiao,  Panpan Wang,  Linsong Dong,  Yaguang Zhang,  Zhaofang Han, Qiurong Wang and  Zhiyong Wang

Fisheries College, Jimei University, Xiamen, Fujian, China

## ABSTRACT

Whole-genome single-nucleotide polymorphism (SNP) markers are valuable genetic resources for the association and conservation studies. Genome-wide SNP development in many teleost species are still challenging because of the genome complexity and the cost of re-sequencing. Genotyping-By-Sequencing (GBS) provided an efficient reduced representative method to squeeze cost for SNP detection; however, most of recent GBS applications were reported on plant organisms. In this work, we used an *Eco*RI-*Nla*III based GBS protocol to teleost large yellow croaker, an important commercial fish in China and East-Asia, and reported the first whole-genome SNP development for the species. 69,845 high quality SNP markers that evenly distributed along genome were detected in at least 80% of 500 individuals. Nearly 95% randomly selected genotypes were successfully validated by Sequenom MassARRAY assay. The association studies with the muscle eicosapentaenoic acid (EPA) and docosahexaenoic acid (DHA) content discovered 39 significant SNP markers, contributing as high up to ∼63% genetic variance that explained by all markers. Functional genes that involved in fat digestion and absorption pathway were identified, such as *APOB*, *CRAT* and *OSBPL10*. Notably, *PPT2* Gene, previously identified in the association study of the plasma n-3 and n-6 polyunsaturated fatty acid level in human, was re-discovered in large yellow croaker. Our study verified that *Eco*RI-*Nla*III based GBS could produce quality SNP markers in a cost-efficient manner in teleost genome. The developed SNP markers and the EPA and DHA associated SNP loci provided invaluable resources for the population structure, conservation genetics and genomic selection of large yellow croaker and other fish organisms.

Corresponding author
Zhiyong Wang, zywang78@qq.com, zywang@jmu.edu.cn

## INTRODUCTION

Whole-genome single nucleotide polymorphism (SNP) is one of the most important genomic resources for population diversity, conservation genetics and functional gene identification for biological traits (*Seeb et al., 2011*). To obtain the molecular markers of the shared genomic loci among individuals, many technologies were invented and developed to probe whole-genome polymorphisms. The techniques allowing synthesizing DNA probes in chips have led to the advent and application of SNP microarrays (*Lipshutz et al., 1999*), making it possible to explore genome-wide SNP in a high-throughput manner. However the cost of array design and application obstructs the wider usage in non-model species, especially for endangered and economic organisms (*De Donato et al., 2013*). More importantly, microarray approaches cannot discover novel SNP loci for species without reference sequences (*Popova et al., 2013*). With the development of next-generation sequencing (NGS), the state-of-art sequencing platform enable scientists to scan small variants in genomes at an unprecedentedly scale with rapidly decreasing price. The multiplex library strategies were widely used to further reduce the cost per sample. However, the budget is still one of the biggest challenges for whole-genome resequencing in non-model samples (*Muir et al., 2016*). Furthermore, the whole-genome sequencing data for hundreds of individuals also inevitably burdens the limited computational and bioinformatics capacity in labs.

In the past few years, several robust sequencing-based genotyping techniques have been invented in the research community to overcome the bottle-neck of cost in whole-genome resequencing. Most of those innovations employ a strategy of partial genome representation sequencing (*Narum et al., 2013*), such as restriction site associated DNA (RAD) (*Rowe, Renaut & Guggisberg, 2011*), IIB restriction endonucleases based RAD (2bRAD) (*Wang et al., 2012*) and Genotyping-By-Sequencing (GBS) (*De Donato et al., 2013*). RAD applies a restriction enzyme to digest genome DNA and then random fragment them to generate RAD tags. Although RAD experiments was initially designed for microarray-based genotyping (*Miller et al., 2007*), the updated RAD tag isolation and library construction procedure has been prevalently used to couple with high-throughput sequencing on the Illumina platforms, resulting many successful applications for genome-wide genotyping, genetic mapping, quantitative trait locus (QTL) and association studies (*Baird et al., 2008*). However, RAD still depends on random fragmentations, reducing the consistence on SNP loci among samples. *Elshire et al. (2011)* subsequently developed a more straightforward genotyping method as GBS with restriction enzymes of *Ape*KI in maize and barley. The protocols for GBS are simple, extremely specific and highly reproducible. In recent years, the easy transferability of GBS to other species leads to many application in plants (*Poland & Rife, 2012*). One of the most attracting features of GBS is the using of methylation-sensitive restriction enzymes during libraries constructions to avoid repetitive fragments and to simplify the reads alignments in extremely complex genomes (*Elshire et al., 2011*); therefore, GBS is an excellent whole-genome genotyping technique for complex non-model organism genomes with massive repetitive regions and abundant genetic diversities.

Teleost, representing a large portion of fish species, has been showed to undergo the third round of whole-genome duplication (WGD) 370 million years ago (*Braasch et al., 2016*; *Xiao et al., 2015b*). The extra WGD left a large portion of duplicated and repetitive sequences in teleost genomes (*Berthelot et al., 2014*; *Jaillon et al., 2004*), making the accurate whole-genome SNP marker development was still challenging in many teleost species (*Wang et al., 2008*). We speculated that GBS technique provided an efficient way and was suitable for genotyping in teleost complex genome. However, the whole-genome SNP development and association studies based on GBS is rarely reported on teleost fish species. Large yellow croaker (*Larimichthys crocea*), belonging to the Sciaenidae family of teleost, is an important marine fish in China and East Asia (*Xiao et al., 2015a*). Due to over-fishing and habitat degradation in last decades, the wild stock of the species has rapidly collapsed (*Liu, Mitcheson & Sadovy, 2008*). The environmental changes and over-dense aquaculture pose more challenges on population conservation and sustainable development of the aquaculture for large yellow croaker. Whole-genome molecular markers and genome-wide association studies (GWAS) for important traits are prerequisites for the population conservation and genomic selection of the species (*Steiner et al., 2013*). However, the association studies are rarely reported for large yellow croaker, largely because of the lacking of abundant stable genomic SNP markers.

GBS technique provides the potential cost-efficient way for whole-genome SNP marker development in complex teleost genome. In the present investigation, we used large yellow croaker to verify the applicability of GBS on teleost. Two restriction enzymes of *Eco*RI and *Nla*III based GBS protocol was developed and optimized. Massive whole-genome SNP markers were developed from the sequencing reads by bioinformatic pipelines, which were subsequently validated by Sequenom MassARRAY assay. The detected SNP markers in this work were then applied to the whole-genome association study of the muscle Eicosapentaenoic Acid (EPA) and Docosahexaenoic Acid (DHA) content in large yellow croaker. Our study confirmed the suitability of GBS on whole-genome SNP marker development in teleost genome. The developed whole-genome SNP markers and functional genes involved in muscle EPA and DHA contents offered valuable genetic resources for conservation genetics and genomic selection of large yellow croaker.

## MATERIALS AND METHODS

### Ethics statement

The sample collection and experiments in the study was approved by the Animal Care and Use Committee of Fisheries College of Jimei University (Animal Ethics no. 1067).

### Sample preparation and DNA extraction

The mixed reference population of 500 individuals was bred by a random fertilization of 30 males and 30 females at the large yellow croaker breeding base of Jimei University in Ningde, Fujian, China. All fish individuals were 1.5 year old with the total length and weight of 24.5–25.9 cm and 217.8–234.1 g (95% confidence interval), respectively. To extract genomic DNA respectively from 500 individuals, the dorsal fins (20–30 mg) of the fish individuals were collected, frozen in liquid nitrogen for the following DNA extraction.

Total genomic DNA was prepared in 1.5 ml microcentrifuge tubes containing 550 µl TE buffer (100 mM NaCl, 10 mM Tris, pH 8, 25 mM EDTA, 0.5% SDS and proteinase K, 0.1 mg/ml). The samples were incubated at 55 °C overnight and subsequently extracted twice using phenol and then phenol/chloroform (1:1) method. DNA was precipitated by adding two and a half volumes of ethanol, collected by brief centrifugation, washed twice with 70% ethanol, air dried, re-dissolved in TE buffer (10 mM Tris–HCl, 1 mM EDTA, pH 7.5). DNA concentration and quality were estimated with an ND-1000 spectrophotometer (NanoDrop, Wilmington, DE, USA) and by electrophoresis in 0.8% agarose gels with a lambda DNA standard.

### *In silico* enzyme assessment for GBS library construction

To assess how endonucleases influence the DNA fragment length distribution, four enzyme combinations were designed for *in silico* digestion of large yellow croaker genome: *Ape*KI-*Pst*I, *Eco*RI-*Bst*NI, *Eco*RI-*Nla*III and *Pst*I-*Nla*III (NEB, Ipswich, MA, USA). The activation and heat-inactivation temperature, location in chromosome and length distribution of all fragments were analyzed to evaluation the best performance of each enzyme combination.

### GBS library construction and sequencing

The GBS libraries were constructed based on two DNA endonucleases: *Eco*RI (NEB, Ipswich, MA, USA) and *Nla*III (NEB, Ipswich, MA, USA). A pilot GBS experiment was performed before the library construction to optimize the temperature and time parameters for yield, size distribution. Based on the pilot experiment, the GBS libraries of large yellow croaker based on *Eco*RI and *Nla*III were constructed following the similar method in previous report (*Beissinger et al., 2013*). Briefly, genomic DNA (20 ng/µl) was incubated at 37 °C with *Eco*RI and *Nla*III, 10XCutSmart$^{TM}$ Buffer. The restriction reactions were heat-inactivated at 65 °C by 20 min and were kept in 8 °C for the following experiments. Sequencing adaptor and barcode mix, T4 DNA Ligase, 10 mM ATP and 10XCutSmart$^{TM}$ Buffer were incubated at 16 °C for 2 h for ligation reactions. The reactions were then heat-inactivated at 65 °C by 20 min and the reaction systems were kept in 8 °C. Then, polymerase chain reactions (PCR) experiments were performed in the reaction solutions (20 µL) containing the diluted restriction/ligation samples (4 pM, 2 µL), dNTP (each at 10 mM, 5 µL), Taq DNA polymerase (NEB, Ipswich, MA, USA) (5 units/µL, 0.25 µL), Illumina Primers (each at 10 µM, 1 µL) and Indexing Primers (10 µM, 1 µL). The PCR procedure was: 95 °C 2 min; 15 cycle of 95 °C 30 s 60 °C 30 s, 72 °C 30 s; 72 °C 5 min and kept in 4 °C. The PCR products were run on a 8% polyacrylamide gel electrophoresis. Fragments of 200–300 bp were isolated using QIAGEN QIAquick$^®$ Gel Extraction Kit and diluted for pair-end sequencing on an Illumina HiSeq 2500 sequencing platform (Illumina, Inc, San Diego, CA, USA).

### Sequencing read quality control and genotyping

The raw sequencing reads generated by Illumina HiSeq 2500 from the GBS libraries were treated and cleaned for SNP detection. First, the adaptors were removed and the resulted reads were split by sample-specific barcode sequences. Only reads begins with

the digest site sequences of *Eco*RI and *Nla*III were retained for the following quality control. Second, the overall base and read quality were accessed by FastQC. To avoid the negative influence of ambiguous bases for SNP detection, reads with more than 5% of N were removed. Then, the resulted reads were cleaned by the following steps: (1) discarding the reads that the quality lower than 20; (2) deleting 5 bp windows in reads end that the average quality smaller than 20; (3) removing read pairs if one end was shorter than 50 bp.

The cleaned reads were mapped to large yellow croaker genome by BWA 0.7.6a (*Li & Durbin, 2009*). The mapping was preceded by a short reads alignment with BWA-MEM algorithm. The alignment were then sorted by coordinates and duplicate marked by SortSam and MarkDuplicates programs in Picard tools 1.107 (picard.sourceforge.net), respectively. To reduce the false positives of SNP detection in this study, three processes were carried out: (1) short read mapping were re-aligned by local bases matches; (2) base Quality Score Recalibration (BQSR) was employed to adjust the accuracy of the base and mapping quality scores; (3) only reads pairs that both aligned on genome with a mapping score higher than 30 were used for SNP calling. Then, the SNP markers were detected by GATK UnifiedGenotyper utility.

## SNP validation by Sequenom MassARRAY assay

Genomic DNA was extracted from dorsal fin ray tissue as the method described before. PCR amplification was performed in the reaction system (5 µl total volume) containing 20 ng of genomic DNA, 0.5U HotstarTaq (Qiagen), 0.5 µl 10× PCR buffer, 0.1 µl dNTPs for each nucleotide and 0.5 pmol of each primer. All PCR experiments were carried out in a PTC-100 PCR instrument (Eppendorf) with the following program: 4 min denaturation at 94 °C, 35 cycles of 20 s at 94 °C, 30 s at 56 °C and 1 min at 72 °C and a final extension at 72 °C for 3 min. After the PCR products were cleaned using 2 µl SAP (SEQUENOM), the single base extension used 2 µl EXTEND Mix (SEQUENOM) contained 0.94 µl Extend primer Mix, 0.041 µl iPLEX enzyme and 0.2 µl iPLEX termination mix and performed with the following steps: initial denaturation at 94 °C for 30 s, followed by 40 cycles of 3-step amplification profile of 5 s at 94 °C, additional 5 cycles of 5 s at 52 °C and 5 s at 80 °C and a final extension at 72 °C for 3 min.The PCR product was cleaned by resin purification and then analyzed using MassARRAY Analyzer Compac (SEQUENOM) and software TYPER (SEQUENOM).

To evaluate the accuracy of SNP detection in this study, the genotypes from GATK SNP calling were compared with those from MassARRAY assay. If the genotypes of one SNP locus from GATK calling were identical with that in MassARRAY, then the locus was called a correct genotype. As a result, 1,421 of 1,500 SNP loci were correctly genotyped by GATK and the success rate of SNP calling was ∼94.7%. The specificity and sensitivity of SNP calling in the study were also evaluated. The reference homozygous genotypes (AA) both from MassARRAY and GATK were called true negatives, and the heterozygous genotypes or allelic homozygous (AB and BB) both from MassARRAY and GATK were called true positives. Specificity was then calculated as the number of true positives divided by the number of true positives plus the number of false positives, and the sensitivity was estimated

as the number of true positives divided by the number of true positives plus the number of false negatives as the following formula:

$$\text{Specificity} = \frac{\text{true negative}}{\text{true negative } + \text{ false positive}}$$

$$\text{Sensitivity} = \frac{\text{true positive}}{\text{true positive } + \text{ false negative}}.$$

## Association analysis with the muscle EPA and DHA content

From the 500 large yellow croaker population, 200 individuals were randomly selected for the muscle EPA and DHA content measurement for the following statistics and association analysis. The fat acid composition analysis followed the similar methods in previous reports (*Murillo, Rao & Durant, 2014*). Briefly, the total lipid was extracted from the fresh muscle tissue using the chloroform-methanol method (*Folch, Lees & Sloane-Stanley, 1957*). After saponification with 1 ml of 50% KOH in 15 ml ethanol, the lipid was then esterified in 80 °C for 20 min using 6.7% boron trifluoride (BF3) in methanol (Morita chemical industries Co., Ltd., Osaka, Japan). After making up in hexane (20 mg/ml), fatty acid methyl esters (FAME) preparations were analyzed by gas chromatography (GC). The temperature increase of 170–260 °C at 2 °C/min was set and helium was used as the carrier gas. Since the muscle contents of EPA and DHA were highly correlated, we combined those two components together in fish muscle.

Prior to the association study, pairwise clustering based on the alleles shared identical by state (IBS) between any two individuals was performed to assess the genetic relatedness. The population stratification was analyzed by multi dimensional scaling (MDS) clustering methods available in the PLINK software (*Purcell et al., 2007*).

With the developed SNP markers, the association analysis was performed between genotypes and measured muscle EPA and DHA content using Plink 1.07 A simple linear regression of phenotype on genotype was performed in the analysis. Markers with $p$-values $\leq$ 1e–4 were considered significantly associated with muscle EPA and DHA contents. To identify the biological functions of nearby genes and whether the orthologs of these significantly associated loci were also associated with the EPA and DHA content in other species, we identified the protein-coding genes around 50 kb of the significant SNP markers. We aligned the genes against the NCBI nr database by Blastx (*Altschul et al., 1997*). GO term and KEGG pathway enrichment analysis of the associated genes were performed with Gene set enrichment analysis (GSEA) (*Shi & Walker, 2007*) by two-tailed Fish's exact test with Benjamini & Hochberg false discovery rate (FDR) (*Benjamini & Hochberg, 1995*) against the background of the all protein-coding gene in large yellow croaker genome. The additive genetic variances were estimated by using R-package EMMREML, Version 3.1. (http://mirror.bjtu.edu.cn/cran/web/packages/EMMREML/index.html).

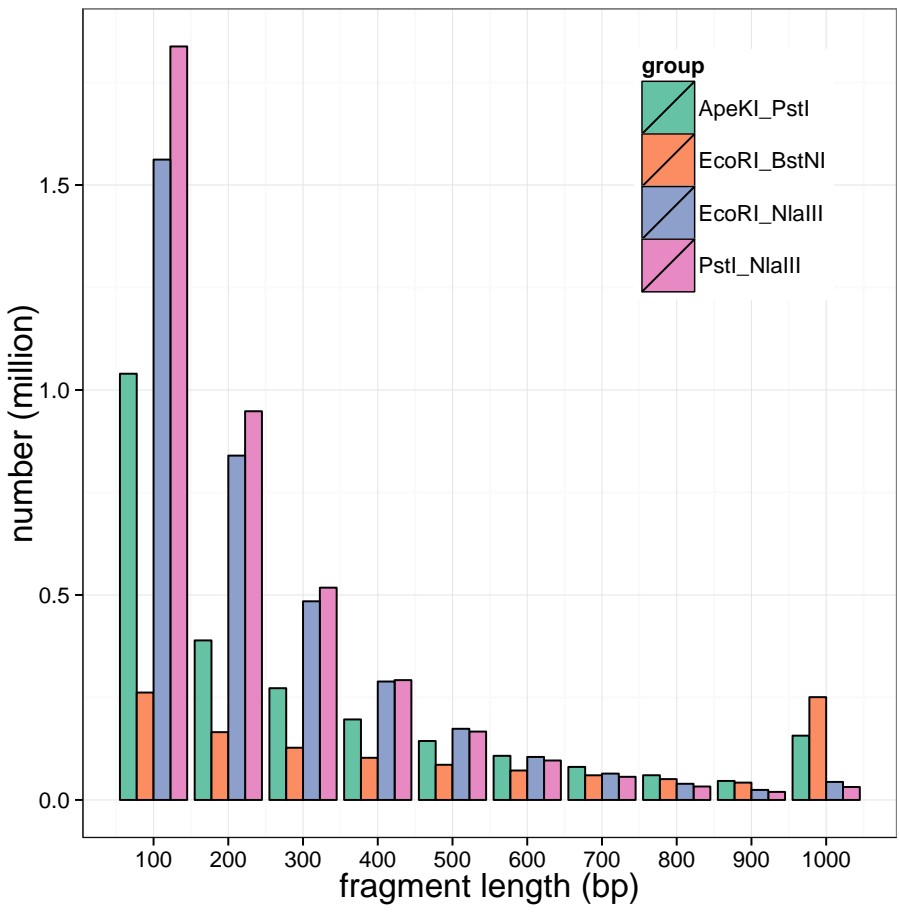

**Figure 1** **Fragment length distribution by restraint enzyme combination.** Note that all fragments longer than 1 kb were accumulated in the last bar.

## RESULTS

### Enzyme assessment and GBS construction for large yellow croaker

According to the principles of the enzyme combination design for GBS library construction, four enzyme combinations were designed for the GBS analysis of large yellow croaker genome: *Ape*KI-*Pst*I, *Eco*RI-*Bst*NI, *Eco*RI-*Nla*III and *Pst*I-*Nla*III (NEB, Ipswich, MA, USA). To assess the fragment size distribution and the number of potential SNP marker developed, the public large yellow croaker draft genome sequences (*Ao et al., 2015*) were *in silico* digested by the four two-enzyme combinations to mimic the genomic fragmentation. As shown in the Fig. 1, the predicted fragment numbers decreased with the fragment size for all enzyme combinations, but the *Ape*KI-*Pst*I and *Eco*RI-*Bst*NI lead to a large portion of fragments longer than 1 kb. According to the size distribution in Fig. 1 and to make the fragment size more compatible to NGS sequencing, the genomic fragment with a size of 100–300 bp were preferred; therefore, *Eco*RI-*Nla*III and *Pst*I-*Nla*III were the rational combinations for the following library construction. According to our *in silico* experiments, genomic fragment in the range of 200–300 bp were used to construct GBS libraries (see

'Method' for details). By the assessment of the combination of *Eco*RI-*Nla*III, roughly 1.5 million fragments would be collected in libraries.

## Library sequencing and reads mapping

GBS libraries were constructed by the two enzyme based digestion (see 'Method' for the details). The NGS sequencing of GBS libraries for 500 individuals generate roughly 314 Gb raw sequencing reads. To evaluate the raw data distribution among samples, we found that the majority of individuals (∼95%) had the raw sequencing reads ranged from 600 to 650 Mb, indicating the excellent sequencing uniformity among samples from library construction and sequencing. The raw reads were cleaned by HTSeq to trim low quality ends (average quality < 20) and eliminate short reads (length < 50 bp). The cleaned reads were mapped to large yellow croaker reference genome sequences (*Ao et al., 2015*) by BWA (*Li & Durbin, 2009*). To assess the quality of GBS library, the mapped reads distribution of Sample 88 along the linkage groups were illuminated as an example (Fig. S1). We found that reads were evenly covered all linkage groups of large yellow croaker, indicating an ideal representativeness of the libraries at the whole genome level. The covered loci depth distribution (Fig. S2) showed that the majority of depth ranged from 5 to 20 and extreme reads enrichment on genome local regions were successfully avoided in sequencing libraries.

## SNP discovery among samples in large yellow croaker genome

To develop molecular markers based on the GBS library sequencing, SNP variants markers were detected from the reads alignments by GATK (*McKenna et al., 2010*) pipelines (see 'Method and Material' for detailed information). To improve the quality of the detected SNP, we employed the extra reads local re-alignment and Base Quality Score Recalibration (BQSR) steps in SNP calling pipelines. Previous literatures on model organisms showed that those extra processes on reads alignment and SNP quality could significant reduce the false positives SNPs (*De Pristo et al., 2011*; *McKenna et al., 2010*; *Van der Auwera et al., 2013*), therefore our refined bioinformatics pipeline coupled with the library construction provided a solid foundation for SNP detection in this study. As a result, 489,246 SNP markers were discovered in at least 200 large yellow croakers with a loci depth threshold of 3. It is not surprised that the majority of SNP markers were not shared by all samples because of the inherent DNA polymorphisms on enzyme digestion site in genome. The number of shared SNP markers among samples was crucial for the evaluation of the GBS sequencing of large yellow croaker genome, especially for the studies of QTL and GWAS analysis in populations. We further used depth- and population-based method to investigate the influence of loci depth and population size on the number of shared SNP marker. As we expected, both the population size and loci depth dramatically influenced the number of shared SNP markers (Fig. 2). However, hundreds of thousands of the shared SNP markers were identified with a depth threshold of 5 in the study. According to previous literatures on SNP development in non-model organisms, the depth filtering of 5 provided high quality SNP markers for the genetic studies (*Hiremath et al., 2012*; *Nguyen, Hayes & Ingram, 2014*); therefore, our SNP calling based on GBS library developed sufficient SNP markers for the biological trait mapping and conservation genetics. We indeed found the

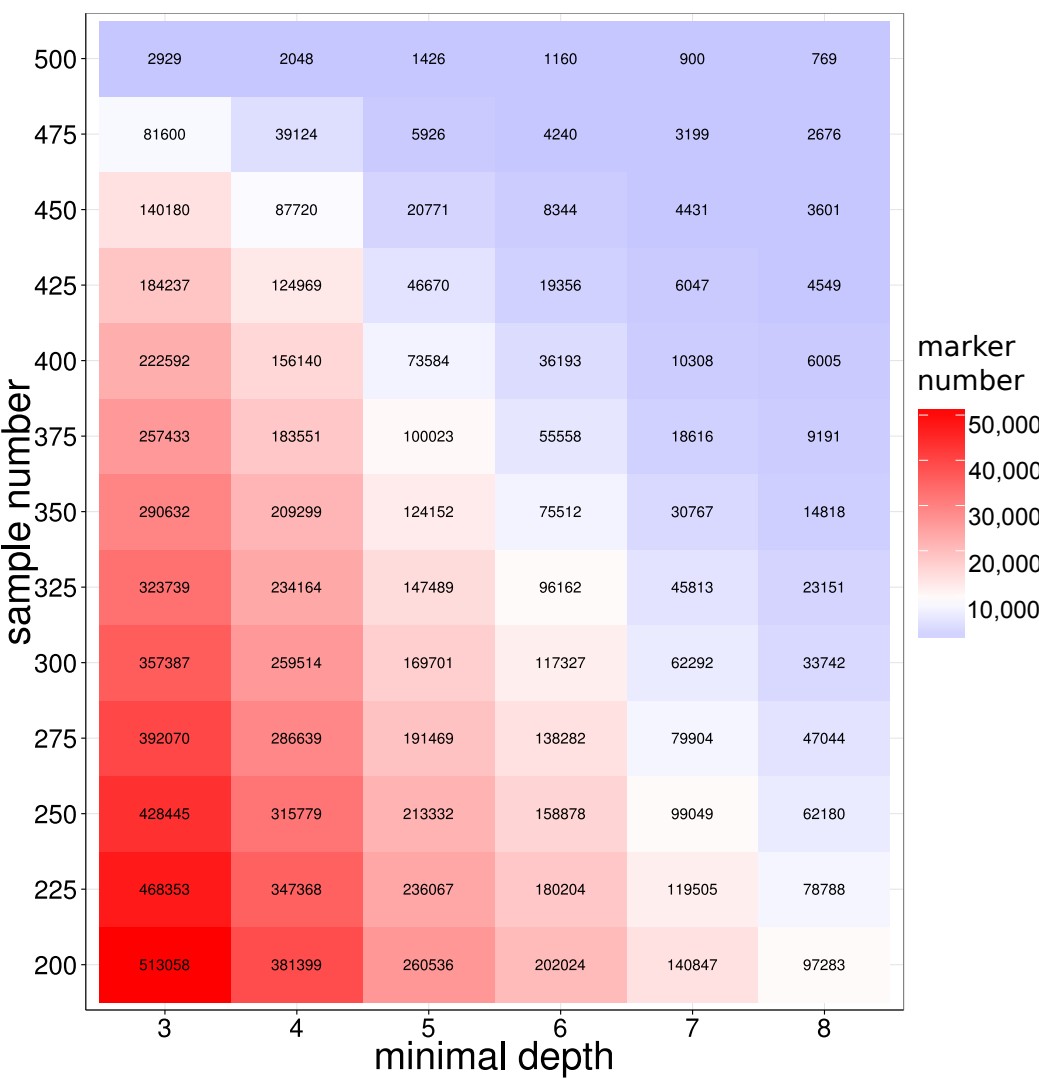

**Figure 2 SNP number against sequencing depth and completeness.** Note that only SNP loci witha quality score larger than 100 were used in the analysis.

sharp decreases on the number of shared SNPs for the population size from 450 to 500, which could be attributed to the samples with extremely low sequencing amount.

To control SNP marker quality while maximizing the number of shared samples and to facilitate the following GWAS analysis, markers with the loci depth higher than 5 and the shared in at least 400 individuals (90% of all sample) were used for the following analysis, resulting 69,845 SNP markers in large yellow croaker genome. To answer the question if our sequencing data was sufficient for the whole-genome SNP development, the numbers of the detected genomic SNP markers were plotted against the sequencing data for each sample. As shown in Fig. 3, the number of the discovered SNP marker increased with sequencing reads and remained to be ~70,000 when the sequencing amount reached 600 Mb, implying that 600 Mb might be an optimal data amount for the trade-off the cost and SNP number in our large yellow croaker GBS libraries. The distribution of those SNP

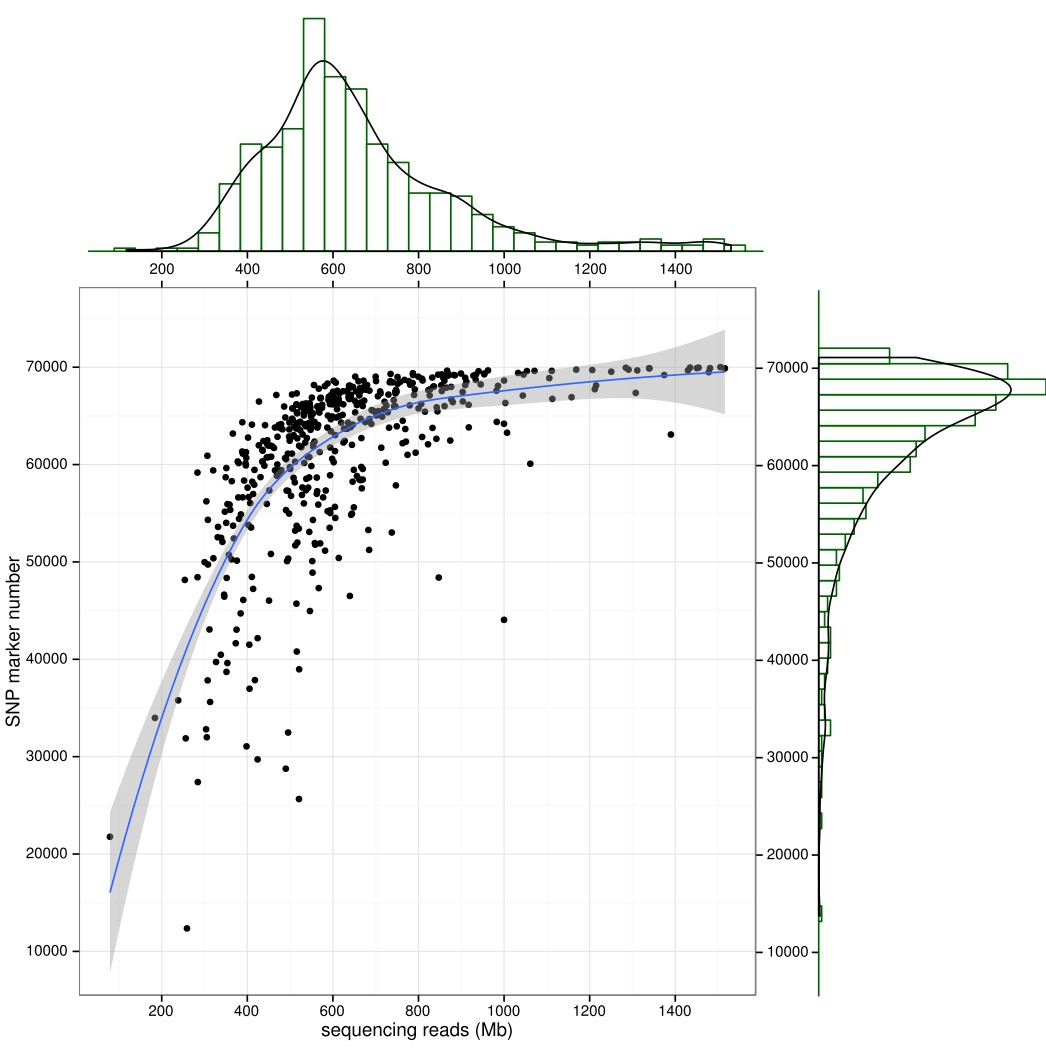

**Figure 3** **SNP number against sequencing amount.** The distribution of sequencingamount (top) and SNP marker number (right) were plot by sides. The line in the scatter is the smoothed curve cross all samples, and the grey area represent the 95% of the confidence region.

markers in 24 linkage groups (Fig. 4) showed that those SNP markers were ideally evenly distributed in the genome, suggesting an excellent representation of whole genome markers in large yellow croaker. The location and functions of SNPs were investigated by comparing the locus coordinates with those of gene annotations. We found that ∼53 % of these SNPs were from genic regions, including exons (3,000), introns (27,114), and untranslated regions (UTRs, 9,166) (Fig. 3). The detailed SNP categories in UTR revealed that 4,311 and 4,855 SNPs were from 5UTR and 3UTR respectively. The biological functions of those SNP markers were analyzed according to their relative positions of the protein-coding genes. As in Fig. 5, 866 SNPs markers in coding regions caused synonymous mutations. Of the remaining markers, 3,022 SNPs could lead to a change of amino acid and introduction of frame shift and new or lost start/stop codons. Those SNP markers might significantly alter

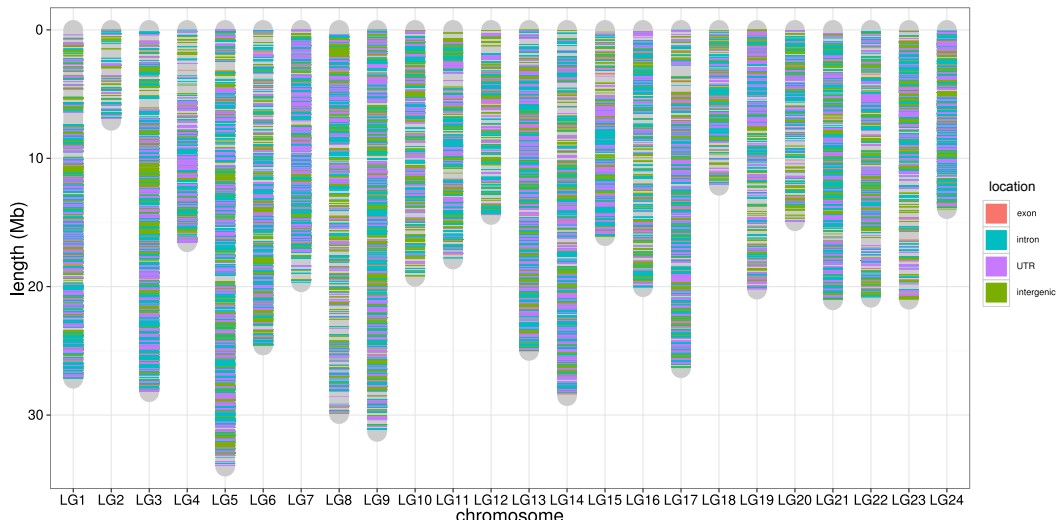

**Figure 4  SNP distribution along chromosome.** The lines along the chromosomes represent SNP loci. The SNP location in exon, intron, UTR and intergenic regions are showed by red, blue, purple and green, respectively.

the biological functions of the hosting genes and thus influence the biological traits that controlled by those genes.

## Experimental validation of detected SNP loci

To assess the reliability of the SNP makers developed from the reduced representation libraries, 50 loci from 30 individuals were randomly selected to validate the marker polymorphism by the Sequenom MassARRAY assay. As shown in Table 1, MassARRAY assay verified the most of detected SNP markers in those samples. Among 1,500 markers, 1,421 were validated by MassARRAY, confirming our library construction, sequencing and SNP marker calling pipelines. The primers for SNP validation and the detailed genotypes were listed in Tables S1 and S2, respectively. As shown in the Table 1, the specificity and sensitivity for the SNP genotype detection in the present study were estimated as 94.2% and 98.3%, respectively. Notably, we found that the majority of discordant genotypes were heterozygous, which was consistent with the reports for other organisms (*Sonah et al., 2013*). We attributed the error-prone genotypes in heterozygous markers to the fact that those markers need more supporting reads than their homozygous counterparts. However, the Sequenom MassARRAY assay still successfully validated ∼95% of the detected SNP marker developed by the GBS library sequencing, providing us solid SNP genotypes of the following trait association and other genetics studies for large yellow croaker.

## The association study with the muscle EPA and DHA content

To apply the genome-wide markers to probe potential marker and genes contributing to muscle EPA and DHA contents, 200 large yellow croakers reared with the identical feed in the same netcage were used to quantify EPA and DHA level. Muscle EPA and DHA contents in 176 individuals were successfully extracted and measured. The contents exhibited a typical normal-like distribution (*p*-value of 0.94 with Kolmogorov–Smirnov

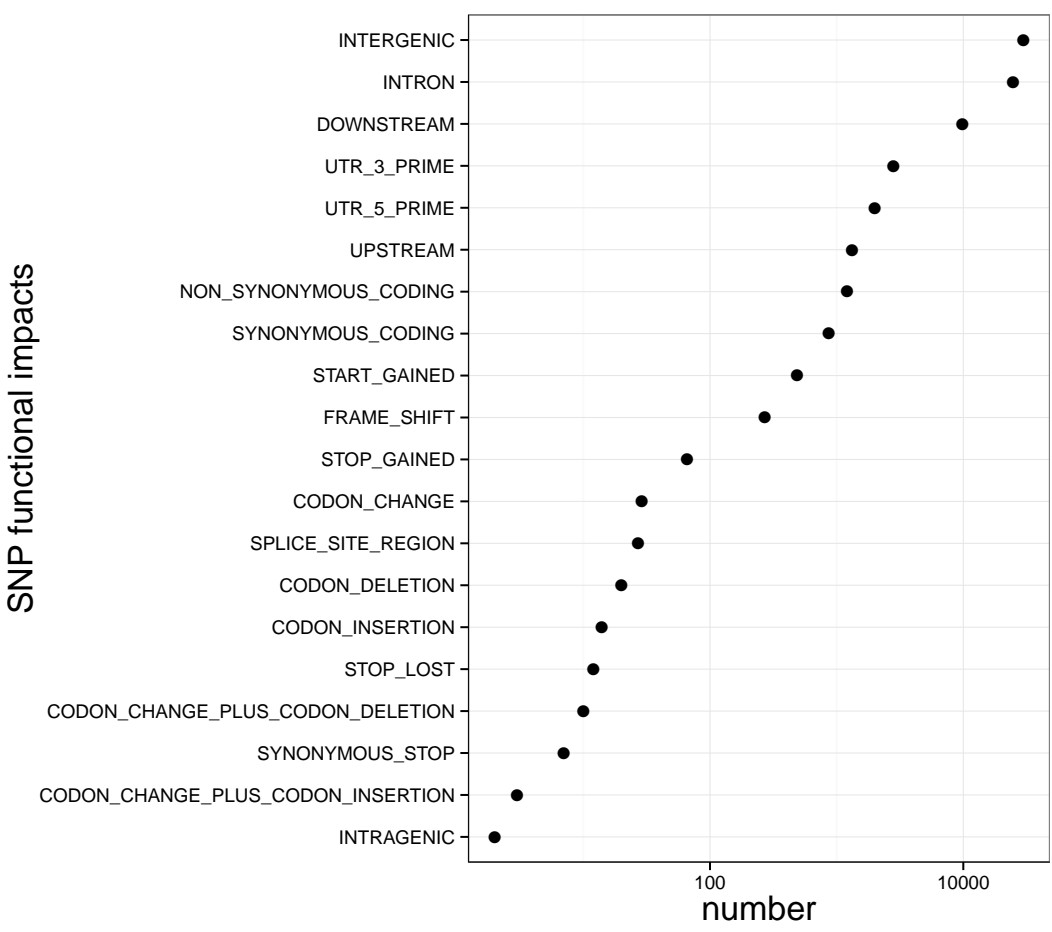

**Figure 5 Biological impact annotations of high quality SNP markers that shared by at least 80% of the population with 500 large yellow croakers.**

**Table 1 SNP validation by Sequenom MassARRAY assay.** NNs indicate the failed genotypes during the SNP filtering.

| Genotypes | | Sequenom MassARRAY assay | | | |
| --- | --- | --- | --- | --- | --- |
| | | AA | AB | BB | NN |
| SNP calling | AA | 901 | 0 | 2 | 0 |
| | AB | 54 | 404 | 7 | 0 |
| | BB | 0 | 2 | 116 | 0 |
| | NN | 11 | 1 | 2 | 0 |

test) with an average of 21.5 mg/g and a standard deviation of 4.1 mg/g (Fig. S3). The difference of the highest and the lowest EPA and DHA contents was ~13.8 mg/g.

Before the association study, population stratification with MDS clustering showed that all samples were grouped into several clusters (Fig. S4). 200 individuals used in the association study were distributed among population clusters, providing diversified genetic background for the following studies. From the MDS plot (Fig. S4), the first and second

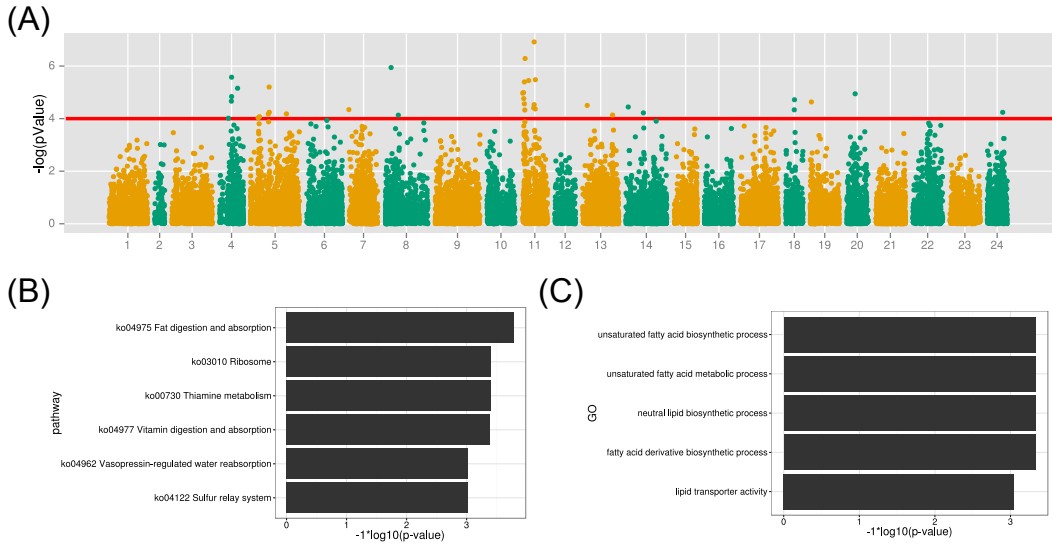

**Figure 6** **GWAS analysis on the muscle EPA and DHA content and the functional analysis therelated protein-coding genes.** (A) The association results were illuminated in the Manhattan plot. The red line is the *p*-value threshold for significant markers; (B) KEGG pathway enrichment of functional genes; (C) GO term enrichment of related biological functions for the associated genes.

dimension value (arbitrary value) were all smaller than 0.05, indicating homogeneous genotypes among samples. The association study of SNP marker with the muscle EPA and DHA content was performed with the linear model with a covariance to sex in Plink (*Purcell et al., 2007*). 69,845 SNP loci developed above with depth threshold of 5 were used to perform the association study (Fig. 5). As shown in Fig. 6, 39 markers from 11 linkage groups were exhibited significant association with the EPA and DHA content ($p$-value < 1e–4). Notably, many associated markers were significant by clusters in linkage group 4, 5 and 11, suggesting the credibility of the association studies. The results might also imply that many genes contributed to the muscle EPA and DHA levels in large yellow croaker. With the variance estimation by Restricted Maximum Likelihood (REML) method (*Smith & Graser, 1986*), we found that those 39 significant markers could interpret as high up to ∼63.0% of genetic variance explained by all 69,845 markers.

To identify gene contributing to the muscle EPA and DHA content in large yellow croaker, we investigated the biological functions of protein-coding genes within 50 kb of all significant SNP markers ($p$-value < 1e–4). As a result, 122 genes were identified from the above association regions. The biological KEGG pathway and GO term annotations of the associated genes were enriched under the background of all protein-coding genes. The metabolic pathway of fat digestion and absorption was significant (FDR < 0.023) in the KEGG enrichment (Fig. 6B and Table S3). Meanwhile, GO terms of unsaturated fatty acid biosynthetic process, fatty acid derivative biosynthetic process and lipid transporter activity were also highlighted (FDR < 0.05) for the associated functional genes (Fig. 6C). The detailed gene function GO annotations were summarized in Table S4. We found that the many identified genes played important roles in lipid transport, metabolism and transcription regulation, such as apolipoprotein B (*APOB*), Carnitine O-acetyltrasferase

(*CRAT*) and oxysterol binding protein 10 (*OSBPL10*). *APOB* is a crucial lipid transport protein in organism. Previous nutriology studies confirmed the correlation of EPA and DHA contents with APOB genotypes and gene expression (*Anil, 2007*). Given their close relationship, we speculated that the polymorphisms on *APOB* gene might contribute to the EPA and DAH accumulation in large yellow croaker muscle. *CRAT* and *OSBPL10* may also involved in the muscle EPA and DHA content since carnitine and oxysterol were important components and regulators in EPA and DHA synthesis pathways according to previous reports (*Qiu, 2003*; *Rise, Marangoni & Galli, 2002*). Notably, we observed palmitoyl-protein thioesteraes 2 (*PPT2*) (around a maker with a *p*-value of 6.7e−06) as a potential functional gene contributing to muscle EPA and DHA contents. *PPT2* gene was also identified by genome-wide association study on n-3 and n-6 polyunsaturated fatty acid levels in Chinese and European-ancestry populations (*Dorajoo et al., 2015*; *Hu et al., 2016*).

## DISCUSSIONS

The advent and development of NGS have unprecedentedly prompted the application of the whole-genome marker development (*Seeb et al., 2011*). Recently, SNP developments on genomic level were performed in many species, including livestock and fish in agriculture (*Sun et al., 2014*). However, the cost for whole-genome re-sequencing is still one of the largest challenges in genomic marker developments. Based on NGS, GBS generally used multiple endonucleases to obtain the desired genomic length and the number of fragments to squeeze the sequencing cost (*De Donato et al., 2013*; *Elshire et al., 2011*; *Sonah et al., 2013*), thus improving the specificity of marker detection along individuals; however, most of the GBS application were reported for plant genomes. Teleost, representing a large portion of fish species, has been showed to undergo the additional third round of whole-genome duplication (WGD) 370 million years ago. The extra genome duplication led to a large portion of duplicated and repetitive sequences in teleost genomes (*Sémon & Wolfe, 2007*). GBS techniques provided an efficient way to probe polymorphism markers from complex genomes; however, the whole-genome SNP development and association studies based on GBS is rarely reported on teleost fish species. In this work, we used teleost large yellow croaker to verify the applicability of GBS on genomic marker development on teleost species. So far as we know, this is the first GBS implementation in large yellow croaker genome. The developed SNP markers provided useful resources for the following genetic studies, including population structure, conservation and functional gene mapping of important traits of the species. The enzyme combination and GBS protocols used in this study could also be valuable reference for other teleost species.

Our *in silico* experiments mimicked the two enzyme digestion on large yellow croaker genome. After the detail investigation of the enzymes, we chose the combination of *Eco*RI and *Nla*III for GBS protocol for two reasons. Firstly, *Eco*RI and *Nla*III possessed the identical heat-inactivation temperature, which facilitated the pilot studies to optimize the experimental conditions for library construction; secondly and more importantly, *Eco*RI was sensitive when restriction site overlaps methylation sequence of CpG islands, therefore the using of the enzyme would partially avoid the digestion in repetitive regions. Many

previous GBS libraries were constructed with the fragment length 100–300 bp or even wider (*Elshire et al., 2011*; *Sonah et al., 2013*); however, we predicted ∼3 million fragment would be generated in that range. The large number of fragment implied a large amount of sequencing reads to cover those genomic regions, which would increase the unit-cost for the sequencing. To reduce the genomic fragments needed to be sequenced for libraries in this study, we attempted to narrow the length range to 200–300 bp, which was predicted to generate roughly 1.5 million genomic fragment for sequencing.

Taking the SNP frequency of 1 per 1,000 bp (*Pushkarev, Neff & Quake, 2009*), the library sequencing might result into roughly 300 thousand SNP markers along large yellow croaker genome. Our estimation was based on the assumption that all individuals have no mutation on endonuclease digesting site and the read depth were high enough to cover SNP loci. However, because of the divergent genomic background among populations, it is very hard to detect all SNP markers that shared by all individuals. In this work, 489,246 raw SNP markers supported by more than three reads were detected with an average sequencing amount of 600 Mb in at least 200 individual from the 500 large yellow croaker population. To facilitate the following marker association study and breeding practise, previous studies proposed several methods to filter the high quality SNP that shared in more individuals, such as depth-based (*Li et al., 2009*), quality score-based (*Brockman et al., 2008*) and population-based (*Bansal et al., 2010*) manner. In this study, we employed a composite strategy for SNP filtering by simultaneously considering loci depth, marker quality and shared population size. As a result, 69,845 SNP markers were left with a depth higher than 5, quality score higher than 100 and shared with at least 80% individuals (400 large yellow croakers). More than half (∼53%) of those detected quality SNP markers resided in genic regions, enabling us to probe the possible association of trait with the nearby functional genes. We noticed that the percentage of markers in genic regions was higher than that of previous reports in soybean (39.5%) (*Sonah et al., 2013*) but lower than that of sweat cherry (65.6%) (*Guajardo et al., 2015*). Those SNP markers generated from the reduced representation library, especially those in genic regions, provided us an easy and efficient manner to detect genomic small variants and to identify genomic regions related to important traits of large yellow croaker at genomic scale. The detected SNP markers were then validated by Sequenom MassARRAY assay for the randomly selected 50 loci in 30 individuals. Although the success rate (94.6%) was slight lower than that reported in the similar study in soybean (98%) (*Sonah et al., 2013*), the library preparation protocol and bioinformatics pipeline provided us high quality genotypes on those SNP loci in the population for the following association studies.

The successful applications of GWAS have greatly prompted the understanding to the genetic bases of important economic traits and would eventually benefit the artificial breeding and population conservation of non-model species (*Correa et al., 2015*; *Narum et al., 2013*) EPA and DHA are both omega-3 poly-unsaturated fatty acids that important in human physiology (*Swanson, Block & Mousa, 2012*). Previous medical experiments demonstrated their positive effects on depressive symptoms in clinical trials (*Hoffmire et al., 2012*) and the essential functions in brain development (*Brenna & Carlson, 2014*). Marine fish is a main source for human EPA and DHA supplement and nutritional properties

of fish meat are highly dependent on polyunsaturated fatty acid levels; therefore the EPA and DHA content in muscle is one of the important indicants for the meat quality of fish. The genetic bases controlling EPA and DHA accumulation in fish species are highly interconnected and not fully revealed. Identifying key SNP loci and functional genes will increase our knowledge of molecular mechanism of polyunsaturated fatty acid synthesis and metabolism in marine fish. To the best of our knowledge, most of the previous researches were focus on the genetic variants on poly unsaturated fatty acid metabolism after fish oil supplements in human or gene expression and EPA and DHA level changes with different feed in fish (*Gregory et al., 2016*; *Li et al., 2014*; *Li et al., 2013*; *Trushenski et al., 2012*). The association studies aiming to identity potential functional genes contributing to EPA and DHA accumulation in fish meat is rarely reported.

Among 176 individuals that were used to measure the muscle EPA and DHA level, the average muscle EPA and DHA content in the top 20 large yellow croakers (28.4 mg/g) was almost two-fold of that in the lowest 20 ones (14.6 mg/g). Given that those fish were reared in the same cage and fed with the identical feed, there was a great potential to raise the muscle EPA and DHA content in large yellow croaker via genetic improvements. Using the developed quality SNP markers by GBS protocol, 39 SNP markers from 11 linkage groups were observed to be significantly associated with muscle EPA and DHA levels. From the coordinates of gene and SNP loci, 122 protein-coding genes were identified around those significant markers. The functional analysis by homological searching found that many genes were involved in fat metabolism and transport, such as *APOB*, *CRAT* and *OSBPL10*. Unsaturated fatty acid biosynthetic process, fatty acid derivative biosynthetic process and lipid transporter activity and fat digestion and absorption pathway were significantly enriched in GO terms and KEGG pathways for the identified functional genes. Meanwhile, we observed large numbers of genes functions in cellular metabolism, gene expression and translation regulation, which may also play a role in modulating muscle EPA and DHA contents (Tables S2 and S3). Interestingly, we identified the potential functional gene of *PPT2* gene in large yellow croaker that was previously discovered during the whole-genome association of plasma n-3 and n-6 polyunsaturated fatty acid level in Asian and European populations (*Hu et al., 2016*). The *PPT2* gene in the linkage group 5 of large yellow croaker might play a similar function in human and also contribute to the muscle EPA and DHA level. This result suggested that teleost fish and human may shared similar metabolic pathway for the polyunsaturated fatty acid synthesis and accumulation; however biological functions of *PPT2* gene for the muscle EPA and DHA content in large yellow croaker and other vertebrates need further gene functional analysis.

## CONCLUSIONS

Teleost were widely believed to undergo the third round of WGD during the natural evolution; therefore, genomes of many teleost species were characterized by the complexity of high heterozygosity and repeat contents. In this work, *Eco*RI-*Nla*III based GBS protocol was used to develop the whole-genome SNP markers in teleost large yellow croaker. The study verified the applicability of GBS on teleost species and provided useful references for

GBS applications in other fish species. For large yellow croaker, about 70,000 high quality SNP markers, supported by at least 400 individuals in population, were detected from the GBS libraries. Those SNP markers were further experimentally validated by Sequenom MassARRAY assay. The even distribution and diversified biological impacts of those molecular makers confirmed the effect and efficiency of the GBS-based SNP development in large yellow croaker. With muscle EPA and DHA contents from 176 individuals, a genome-wide association study between genotypes and EPA and DHA level were performed. 39 and 122 significantly associated SNP loci and related protein-coding genes were identified. The functional analysis of the related genes confirmed the results of the association study.

For the aspect of molecular resources, our developed SNP markers could be valuable genetic resources for large yellow croaker, and would be used in the following population structure, conservation genetics and the association studies for other important economic traits. The associated results for the muscle EPA and DHA content, namely the significant SNP loci and functional genes, provided us important guidance for the further investigation of genetic bases of the muscle EPA and DHA accumulation in large yellow croaker and would eventually aid the technological development towards the genetic improvement of meat quality via the molecular-aided selection of the species.

### Funding

This work was supported by grants from National Natural Science Foundation of China (U1205122 and 31602207), Natural Science Foundation of Fujian Province (2016J05081), National '863' Project of China (2012AA10A403), and the Foundation for Innovation Research Team of Jimei University (2010A02). The funders had no role in study design, data collection and analysis, decision to publish, or preparation of the manuscript.

### Grant Disclosures

The following grant information was disclosed by the authors:
National Natural Science Foundation of China: U1205122, 31602207.
Natural Science Foundation of Fujian Province: 2016J05081.
National '863' Project of China: 2012AA10A403.
Foundation for Innovation Research Team of Jimei University: 2010A02.

### Competing Interests

The authors declare there are no competing interests.

### Author Contributions

- Shijun Xiao conceived and designed the experiments, performed the experiments, analyzed the data, contributed reagents/materials/analysis tools, wrote the paper, prepared figures and/or tables, reviewed drafts of the paper.
- Panpan Wang and Zhaofang Han analyzed the data, reviewed drafts of the paper.
- Linsong Dong analyzed the data, contributed reagents/materials/analysis tools, reviewed drafts of the paper.

- Yaguang Zhang and Qiurong Wang performed the experiments, contributed reagents/materials/analysis tools, reviewed drafts of the paper.
- Zhiyong Wang conceived and designed the experiments, reviewed drafts of the paper.

## Animal Ethics

The following information was supplied relating to ethical approvals (i.e., approving body and any reference numbers):

This sample collection and experiments was approved by the Animal Care and Use committee of Fisheries College of Jimei University (Animal Ethics no. 1067).

## Data Availability

The sequencing short reads were deposited in the NCBI Sequence Read Archive (SRA): https://www.ncbi.nlm.nih.gov/bioproject/?term=309464.

## Supplemental Information

Supplemental information for this article can be found online at http://dx.doi.org/10.7717/peerj.2664#supplemental-information.

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
