# Peer review of "Whole-genome single-nucleotide polymorphism (SNP) marker discovery and association analysis with the eicosapentaenoic acid (EPA) and docosahexaenoic acid (DHA) content in Larimichthys crocea"

_PeerJ, doi:10.7717/peerj.2664_

## Round 0.1 · original submission · Minor Revisions

Please improve your manuscript according to the suggestions given by the two expert reviewers.

Reviewer 1 ·

Basic reporting

No comments

Experimental design

No comments

Validity of the findings

No comments

Additional comments

The manuscript “Whole-genome single-nucleotide 1 polymorphism (SNP) marker discovery and association analysis with the eicosapentaenoic acid (EPA) and docosahexaenoic acid (DHA) content by Genotyping-By-Sequencing (GBS) in teleost Larimichthys crocea” described SNP markers have potential association with the muscle EPA and DHA content of large yellow croaker. It provides some new data to the investigators in this area. However, to be published on PeerJ, there are some questions need to be clarified.

Major comments:

1. Since the authors tried to discover the SNP marker associated with EPA and DHA content, why don’t use the high EPA or DHA content family?
2. Did the DNA extracted from the 500 individuals respectively? If yes, the authors should clarified it at page 8, “Sample preparation and DNA extraction”.
3. Page8, GBS library construction and sequencing, line 161-169, the concentration of genomic DNA and the template of PCR should be clarified.
4. The results, page 11, line 255-260, the sentence “After the detail investigation of the enzymes, we chose the combination of EcoRI and NlaIII for GBS protocol for two reasons. Firstly, EcoRI and NlaIII possessed the identical heat-inactivation temperature, which facilitated the pilot studies to optimize the experimental conditions for library construction; secondly and more importantly, EcoRI was sensitive when restriction site overlaps methylation sequence of CpG islands, therefore the using of the enzyme would partially avoid the digestion in repetitive regions.” Should be moved to the Discussion section.
5. Page11, the authors did not give any description on the enzyme assessment of “ApeKI-PstI and EcoRI-BstNI” in materials and methods, however, it was descripted in the results. They should be consistent in different section.
6. The authors detected the concentration of EPA and DHA together, does that mean the total concentration of EPA and DHA? Are the biosynthesis pathways similar of the two unsaturated fatty acid? If it is right, please clarify them in other sections and figure legends.
7. The manuscript try to provide the association relationship between SNP markers and EPA and DHA content, however, except some annotated genes in some related pathway, there is no potential SNP markers associated with EPA and DHA content. Therefore, I think the authors should give more analysis and provide the SNP information for this.
8. There are some redundant description in results and discussion, the authors should deleted the discussion in the results.

Minor:
9. Page13, line349,“The association study of SNP marker with the muscle EPA and DNA content”, is that DHA?
10. Page 14, line 353 “with the EPA and DNA content”, DNA or DHA?
11. Page 14, line 359, same as above.
12. Page 28, “…… witha ……" should be “with a”.

·

Basic reporting

The work reported by authors developed a very rich collection of SNP markers by sequencing the reduced representative genome of yellow croaker, a very important fish species in China, and located the quantitative traits of EPA and DHA contents. To my knowledge, this is one of the scarce works on economic fish species. The genotyping method should be popularized in aquaculture studies and the findings are interesting.

Experimental design

I have a major concern for the fish individuals used for association analysis. As stated in the manuscript, authors obtained the individuals from a mixed cross of female and male individuals, 30 each parent. No sentence was there to state the random fertilization of these individuals. Even the fertilization was random, then some individuals of the first filial generation may genetically related. That is to say the possibility of non-independence of association analyzing population is high. In case of non-random fertilization, the genetic stratification will be high. Such scenario may interfere the association analysis results. A genetically unrelated population is highly appreciated for association analysis.
This is only my personal concern. Actually, authors may conduct a cluster analysis to reveal the genetic relationship among individuals as their genotypes are known.

Validity of the findings

The authors obtained a rich collection of SNP marker for this important fish species, mapped two important economic traits and revealed the possible genetic and biochemical bases of these two traits. These findings are new and may serve as references of similar studies.

Additional comments

Generally, I appreciate what the authors have down. I suggest its acceptance for publication in Peer J. However, many English writing errors should be corrected ahead of publication.

For example,
Xiamen, the city where authors conducted their work should be followed by post code, a six numbered code in China;
GBS should be fully spelled when it appeared first in abstract;
Line 76, we know multiple PCR method, thus library multiplex strategy should be changed into multiple library construction strategy;
Line 140, committee should be Committee;
Line 168-170, The PCR procedure was: 95°C 2 min; 15 cycle of 95°C 30 sec, 60°C 30 sec, 72°C 30 sec; 72°C 5 min and kept in 4°C. Please find a reference to rewrite this procedure, before and after the word "and", cannot be placed together’
Line 195, 0.1μl dNTPs, 0.1μl dNTP (each nucleotide) ?
Line 217, Association analysis with the muscleEPA/DHA content, a blank should be inserted between muscle and EPA/DHA;

In reference section, capital letters should be there for many of the name of journals; line 623, de should be separated with Pristo; many places, scientific name (Latin name) of a species should be Italicized; for a book, publisher and publishing place should be provided for tracing; why doi number was provided for only a few references? The title of papers should be in the same style, why some of the titles show large letters for the first letter of words? Each article for journals like British Journal of Nutrition and Scientific Reports should have been numbered each volume, why not you cited?

---

## Round 0.2 · accepted · Accept

The paper has been modified and improved and now is ready for publication.